# Research on Underdetermined DOA Estimation Method with Unknown Number of Sources Based on Improved CNN

**DOI:** 10.3390/s23063100

**Published:** 2023-03-14

**Authors:** Fangzheng Zhao, Guoping Hu, Hao Zhou, Shuhan Guo

**Affiliations:** 1Graduate School, Air Force Engineering University, Xi’an 710043, China; 2Air and Missile Defense College, Air Force Engineering University, Xi’an 710043, China

**Keywords:** convolutional neural networks, source number estimation, underdetermined DOA estimation, uniform linear array, sparse array

## Abstract

This paper proposes a joint estimation method for source number and DOA based on an improved convolutional neural network for unknown source number and undetermined DOA estimation. By analyzing the signal model, the paper designs a convolutional neural network model based on the existence of a mapping relationship between the covariance matrix and both the source number and DOA estimation. The model, which discards the pooling layer to avoid data loss and introduces the dropout method to improve generalization, takes the signal covariance matrix as input and the two branches of source number estimation and DOA estimation as outputs, and achieves the unfixed number of DOA estimation by filling in invalid values. Simulation experiments and analysis of the results show that the algorithm can effectively achieve the joint estimation of source number and DOA. Under the conditions of high SNR and a large snapshot number, both the proposed algorithm and the traditional algorithm have high estimation accuracy, while under the conditions of low SNR and a small snapshot, the algorithm is better than the traditional algorithm, and under the underdetermined conditions, where the traditional algorithm often fails, the algorithm can still achieve the joint estimation.

## 1. Introduction

Array signal processing is the use of sensor arrays to extract signal features to achieve signal parameter estimation, signal enhancement, signal separation and other signal processing operations; it is widely used in a variety of civil and military fields such as radar, exploration, and medical diagnosis [1]. Signal parameter estimation mainly includes source number estimation and direction of arrival (DOA) estimation, etc. Generally speaking, source number estimation is a prerequisite for signal-related parameter estimation and has an important position in array signal processing. The earliest proposed method of source number estimation is the hypothesis testing method [2], which performs a characteristic decomposition of the received signal covariance matrix. Theoretically, the first few larger eigenvalues are equal in number to sources and correspond to the signal eigenvalues, while the remaining eigenvalues correspond to the noise eigenvalues, which are usually divided according to empirically set thresholds. This method requires the source number according to the set threshold, which is susceptible to subjective factors and makes it difficult to accurately distinguish between signal eigenvalues and noise eigenvalues under low signal-to-noise ratio (SNR) or small snapshot conditions. The Akaike Information Criterion (AIC) and the Minimum Description Length criterion (MDL) [3], which are regarded as classical source number estimation methods based on information theory, were successively proposed by Akaike. and Rissanen with the development of information theory. They calculate the maximum likelihood estimate of the unknown parameters of the array signal and introduce a penalty function for correction, constituting the discriminant function and the number of sources that can be estimated [4]. Based on the issues of inconsistent estimation and underestimating under low SNR and small snapshot settings, researchers then made a number of changes to the AIC criterion and MDL criterion; to address the failure of the above algorithms under color noise, the Gerschgorin circle algorithm was proposed [5], which uses the property of orthogonality between the signal eigenvector and the noise eigenvector to separate the Gerschgorin circle corresponding to the signal [6]. However, the algorithm takes the eigenspace of a small one-dimensional covariance matrix, losing one system degree of freedom, and the estimation performance is unstable under high SNR conditions. All the above algorithms are only applicable to uniform linear array (ULA), and the number of sources is smaller than the number of physical array elements. For scenarios where the number of sources is greater than the number of physical array elements, sparse arrays are typically utilized to enable multi-source parameter estimation in situations when the number of sources is more than the number of physical array elements. Sparse arrays have array elements separated by more than half a wavelength; a wider array aperture and more degrees of freedom are produced by larger array element spacing. Co-prime arrays, uniformly sparse arrays, minimally redundant arrays, and nested arrays are examples of common sparse array types. Dong et al., designed a spatial covariance model via spatial smoothing of the coprime array output signal to estimate the source number [7]. However, the method relies too heavily on the predefined spatial grid for sparse reconstruction, which is somewhat different from reality. To achieve joint estimation of the source number and DOA, Izedi F. et al., proposed an improved hypothesis testing method based on an arbitrary array type [8]. However, because DOA is unstable, the estimation probability of the source number is subject to changes. To achieve underdetermined source number estimation, Zhang Y et al., used the spatial smoothing method to build a virtual array [9], but this method resulted in a loss of degrees of freedom. Overall, the use of sparse arrays can improve the degrees of freedom of the array, but the methods for source number estimation and DOA estimation are also more complex [10]. Compared to ULA, sparse arrays suffer from ambiguity, and the blurred angles can seriously interfere with the accurate discrimination of source DOA. Matter et al., removed the blurred angles in a 1D line array by using particle swarm optimization [11]; Yu et al., filtered out the spurious peaks by using a double V-shaped array with a common point [12]. The deblurring techniques mentioned above can increase DOA estimation resolution without increasing the array’s virtual elements. New DOA estimation techniques have emerged as sparse arrays with particular structures, such as minimum redundant arrays, co-prime arrays, and nested arrays. The covariance matrix reconstruction method extends the array aperture by reconstructing an extended covariance matrix from the array properties, which is equivalent to a uniform linear array with more physical array elements. This allows for high-resolution angle estimation using the DOA estimation method for ULA. For example, Pal and Vaidyanathan [13,14] extended the signal and noise subspaces of a co-prime array by rewriting the array flow pattern, which increased the resolution and the number of sources that could be detected. However, this type of method is often reserved for specific types of arrays and is difficult to apply widely. In recent years, with the application of deep learning, DOA estimation methods based on deep learning have emerged [15]. Fast matrix decomposition (eigenvalue decomposition and unitary decomposition) was achieved by Luo and Gohil et al., using deep learning, and DOA estimation was then accomplished using conventional algorithms [16,17]. A convolutional autoencoder was used by Liu Z et al., to classify signals coarsely, and multiple DNNs were then used to achieve high accuracy in each sector [18]. Ge et al., discretized the spatial angle with the aid of a convolutional neural network to turn the angle estimation problem into a classification problem. [19]. The above algorithms often require the number of sources as known data, and the estimation accuracy is limited by the angle discretization.

To address the above problems, this paper proposes a joint estimation method of source number and DOA based on an improved convolutional neural network (CNN), which does not need to estimate the number of sources first and then perform angle estimation. The two output branches of CNN are the number of sources and angle values, respectively, which are not constrained by the array flow pattern. The algorithm is still valid when the number of sources is greater than the number of physical and virtual array elements.

## 2. Signal Model

Consider a linear array model with M elements, array element spacing $d=\left\{d_{i}\right\},\;d_{i}=Z_{i}\cdot \left(\frac{\lambda }{2}\right),\;i=0,1,\ldots ,M-1$, where λ is the signal wavelength and $Z_{i}$ is a positive integer; when $Z_{i}$ is all 1, the array is a ULA, otherwise, the array is a sparse array. When K uncorrelated far-field narrowband signals are incident to the array with $\theta_{1},\;\theta_{2},\ldots ,\theta_{K}$, and the source positions are fixed and the centre frequencies are the same, the guidance vector of the k-th signal is
(1)$$a\left(\theta_{k}\right)={\left[1,e^{\frac{j2\pi d_{1}sin\theta_{k}}{\lambda }},\ldots ,e^{\frac{j2\pi \sum_{i=1}^{M-1}d_{i}sin\theta_{k}}{\lambda }}\right]}^{T},$$

Then, the guidance vector matrix is
(2)$$A\left(\theta \right)={\left[a\left(\theta_{1}\right),a\left(\theta_{2}\right),\ldots ,a\left(\theta_{K}\right)\right]}^{T},$$

The array output signal vector is
(3)$$X\left(t\right)={\left[x_{1}\left(t\right),x_{2}\left(t\right),\ldots ,x_{M}\left(t\right)\right]}^{T}=A\left(\theta \right)S\left(t\right)+n\left(t\right)=\sum_{k=1}^{K}a\left(\theta_{k}\right)s_{k}\left(t\right)+n\left(t\right),$$
where $x_{i}\left(i=1,2,\ldots ,M\right)$ denotes the output of the *i*-th array, $S\left(t\right)={\left[s_{1}\left(t\right),s_{2}\left(t\right),\ldots ,s_{K}\left(t\right)\right]}^{T}$ denotes the signal vector, and $n\left(t\right)$ is the additive noise. In general, the source number is often used as a priori knowledge for DOA estimation and needs to be calculated before DOA estimation. The source number can be determined from the eigenvalues of the covariance matrix R when the array is a uniform linear array. The covariance matrix of $X\left(t\right)$ is
(4)$$R=E\left\{X\left(t\right)X^{H}\left(t\right)\right\}=A\left(\theta \right)R_{S}A^{H}\left(\theta \right)+\sigma^{2}I_{M},$$
where $R_{S}=E\left\{S\left(t\right)S^{H}\left(t\right)\right\}$ denotes the signal covariance matrix, $\sigma^{2}$ denotes the noise power, and $I_{M}$ denotes the unit matrix of order M. When the signals are uncorrelated, the column vectors $a\left(\theta_{k}\right),\;k=1,2,\ldots ,K$ of the guidance vector matrix $A\left(\theta \right)$ are non-linearly correlated with each other, and $R_{S}$ is a nonsingular matrix. Under ideal conditions, the rank of the $A\left(\theta \right)R_{S}A^{H}\left(\theta \right)$ matrix is *K*, and the feature decomposition of R yields
(5)$$R=U\Lambda U^{H},$$
where U is the eigenvector matrix, $\Lambda =diag\left\{\lambda_{1},\lambda_{2},\ldots ,\lambda_{M}\right\}$ is the eigenvalue diagonal matrix; the eigenvalues are sorted in ascending order of size, i.e., $\lambda_{1}\geq \lambda_{2}\geq \ldots \geq \lambda_{M}$, where the (M−K) eigenvalues with small values are called noise covariance eigenvalues and have the following characteristics,
(6)$$\lambda_{K+1}=\lambda_{K+2}=\ldots =\lambda_{M}=\sigma^{2},$$

Therefore, the number of signal sources can be calculated from the number of noise covariance eigenvalues. In fact, there is an error between the covariance matrix calculated from the sampled signal and the true value; the noise covariance eigenvalues are no longer identical [20], i.e.,
(7)$$\lambda_{K+1}>\lambda_{K+2}>\ldots >\lambda_{M}>\sigma^{2},$$

For example, to design an ideal ULA with 6 elements and 3 incident signals, $\theta_{1}=-30{}^{\circ}$, $\theta_{2}=10{}^{\circ}$ and $\theta_{3}=30{}^{\circ}$; feature decomposition is performed, and the eigenvalues of the signal covariance matrix at different SNRs and snapshots are calculated, as shown in Table 1.

Each group’s eigenvalues ($\lambda_{1}$ to $\lambda_{6}$) in Table 1 are normalized as shown in Figure 1.

As can be seen from the Table 1 and Figure 1, both a lower SNR and a smaller number of snapshots will have a greater impact on the eigenvalues, with the impact of a low SNR being the most obvious. When SNRs are less than −10 dB, distributions of eigenvalues seen in Figure 1 are almost linear, so it is difficult to distinguish between larger and smaller eigenvalues; this makes it challenging to derive an accurate source number from the distributions of eigenvalues. In addition, from the above derivation process, it can be seen that when the number of sources is greater than or equal to the number of array elements, the number of sources cannot be judged according to the characteristics of the eigenvalue, and the method fails. For the situation where the number of sources is more than the actual number of array elements, sparse arrays are mostly used to increase the array degrees of freedom. The source number estimation method of sparse arrays first vectorizes the received signal covariance matrix to construct a virtual array model [8],
(8)$$R_{v}=vec\left(R\right)=B\left(\theta_{1},\;\theta_{2},\ldots ,\theta_{K}\right)P+\sigma^{2}vec\left(I_{M}\right),$$
where $P={\left[\sigma_{1}^{2},\sigma_{2}^{2},\ldots ,\sigma_{K}^{2}\right]}^{T}$,$\;\sigma_{i}^{2},\;i=1,2,\ldots ,\mathrm{and}\;K$ denotes the signal power of the *i*-th source; when the signals are independent of each other, $B\left(\theta_{1},\;\theta_{2},\ldots ,\theta_{K}\right)$ can be further expressed as:(9)$$B\left(\theta_{1},\;\theta_{2},\ldots ,\theta_{K}\right)=\left[a^{\times }\left(\theta_{1}\right)⊗a\left(\theta_{1}\right),a^{\times }\left(\theta_{2}\right)⊗a\left(\theta_{2}\right),\ldots ,a^{\times }\left(\theta_{K}\right)⊗a\left(\theta_{K}\right)\right],$$

$B\left(\theta_{1},\;\theta_{2},\ldots ,\theta_{K}\right)$ is the extended virtual array; note that the maximum continuous virtual array element response part length in $B\left(\theta_{1},\;\theta_{2},\ldots ,\theta_{K}\right)$ is L. Then, the L×K dimensional uniform array in $B\left(\theta_{1},\;\theta_{2},\ldots ,\theta_{K}\right)$ is selected as the new array-oriented vector, and P can be regarded as the new signal matrix for a single snapshot; the new covariance matrix is obtained by construction as
(10)$$R_{v}^{\prime }=B^{\prime }\left(\theta_{1},\;\theta_{2},\ldots ,\theta_{K}\right)P+\sigma^{2}I_{L},$$
where P can be regarded as the new signal matrix for a single snapshot. $R_{v}^{\prime }$ is spatially smoothed, i.e., the covariance matrix is averaged over all smoothed sub-array covariance matrices to obtain a full-rank covariance matrix $R^{\prime }$. The eigenvalue decomposition is performed on the covariance matrix $R^{\prime }$ according to Equation (5), and the eigenvalues are ranked from largest to smallest, with the number of the first few larger eigenvalues being the number of sources. When the source number is smaller than the actual number of arrays or the maximum number of consecutive virtual arrays of sparse arrays, DOA estimation can be performed according to subspace algorithms, such as the MUSIC algorithm, ESPRIT algorithm, etc.

## 3. Convolutional Neural Network Model

In the estimation of both source number and DOA, the calculation of the received signal covariance matrix is an important step, and there is a non-linear mapping relationship between the received signal covariance matrix and the source number and DOA [21]. The convolutional neural network (CNN) model proposed in this paper is shown in the following Figure 2.

As can be seen from Figure 1, the CNN model proposed in this paper is a single-input multi-branch output neural network model. The received signal covariance matrix serves as the model’s input, and its output is divided into two branches: one produces the number of sources, the other the DOA of all the sources, and the two outputs can both influence and verify one another. Since the source number is unknown, the number of output angles is uncertain, but the model training process requires the input and output to be fixed-length data, i.e., the data size of the input and output is fixed. In this premise, Output2 in this paper’s model is of fixed length, and its length is the maximum possible number of sources received by the array. When the actual number of received sources is smaller than the length of Output2, it is padded with the value N, which is the value in the non-target region. To avoid indistinguishability, N should preferably have a large difference with the target region boundary.

### 3.1. Convolutional Layers

In convolutional layers, a number of square arrays of fixed size, called filters, are given; these are also called convolution kernels. The size of the convolution kernels corresponds to what is called the perceptual field on the input matrix, and the convolution kernels move from left to right and from top to bottom on the input matrix according to the given step size; the perceptual field performs the convolution operation with the convolution kernels to obtain the output matrix, which can be used as the input matrix for subsequent convolutional layers [22]. When performing a convolution operation, the boundaries of the input matrix are computed less frequently than the elements within the matrix, and the size of the output matrix becomes smaller compared to the input during the convolution operation. In order to preserve the boundary features of the matrix as much as possible [23] and to avoid the problem of obtaining feature vectors that are too small in the subsequent convolution layers, the input data are padded before the convolution layer operation, usually with a padding value of 0. The blue arrows in the convolutional layers section of the diagram indicate convolutional operations, and “padding” indicates boundary padding. The model contains three convolutional layers, and the input data or feature vectors are padded before the first two convolutional operations, but not the third. The model uses the “SAME” mode of padding, i.e., the size of the input does not change before and after the convolution. Since there is no special meaning in the single element of the received data covariance matrix, filling the boundary before each convolution may lead to excessive retention of boundary information, resulting in data redundancy; therefore, no boundary filling is performed before the third convolution.

To increase the non-linear capability of the model, a bias term is added after the convolution operation, and a non-linear function is connected, which is called the activation function [24]. The common activation functions include sigmoid function, Tanh function, ReLU function [25], etc. However, the problem of the first two is that there is a gradient saturation problem, and the convergence speed is slow, while the ReLU function converges quickly and does not have the gradient saturation problem. However, the ReLU function will assign the value to zero in the case of less than zero, so that the neuron can no longer be activated. Based on the above problems, this paper chooses the Leaky ReLU function [26], which is an improvement of the ReLU function, and its expression is
(11)$$f\left(x\right)=\left\{\begin{array}{c} x\;\;x\geq 0\\ ax\;x<0 \end{array}\right.,$$

The value of a is generally small, usually a=0.01. The Leaky ReLU function is chosen to avoid the problem of neurons not being re-activated and has a better performance.

In traditional convolutional neural network models, a pooling layer is connected after the convolutional layer, the purpose of which is to achieve feature dimensionality reduction to reduce the size of the data; however, in practice, the number of array elements is often finite, and usually the number is not particularly large, so the size of the covariance matrix is not very large. When the pooling layer is introduced, whether by maximum pooling or average pooling, etc., it will lead to the loss of the finite data, making it difficult for the features to be fully extracted, while not performing boundary filling before the third convolution operation will reduce the size of the feature vector after convolution, and to a certain extent, can achieve reduced complexity.

### 3.2. Fully Connected Layers

Fully connected layers are tiled structures of many neurons, which convert all the feature vectors obtained from the convolutional layer into a one-dimensional feature vector for classification or regression via this layer. The number of layers of fully connected layers (i.e., depth) and the number of neurons in each layer can effectively improve the non-linear performance and complexity of the model, thus enhancing the learning ability of the model; however, too much depth and too many neurons can easily cause overfitting. On the basis of limiting the depth and the number of neurons as much as possible, this paper uses the dropout method [18], which is schematically shown in Figure 3 below.

During the training process, the neurons in each layer are discarded with a certain probability at each iteration. The probability of each layer can be set individually, and the neurons in the output layer are not discarded. The dropout method is used to reduce the constraints of neurons on the training data and to enhance the generalization of the model. In the model of this paper, the dropout method is used for all fully connected layers except for the output layer of two branches.

In the first two layers of the fully connected layers, the activation functions are all Leaky ReLU functions [26]. The first branch, guided by the blue arrow, implements the estimation of the number of sources and belongs to the multi-classification problem. The activation function of the first layer of this branch is the Leaky ReLU function, and the second layer, the output layer, is the Softmax layer, where the Softmax function is used for the multi-classification problem, in which each classification is mutually exclusive:(12)$$\mathrm{Softmax}\left(x_{i}\right)=\frac{e^{x_{i}}}{\sum_{j=1}^{n}e^{x_{j}}}$$

The loss function is the cross-entropy function, whose expression is
(13)$$L\left(p,q\right)=-\frac{1}{M}\sum_{i=1}^{M}\sum_{j=1}^{N}p\left(x_{ij}\right)log\left(q\left(x_{ij}\right)\right)$$
where M denotes the number of samples, N denotes the number of categories, $p\left(x_{ij}\right)$ denotes the probability that the prediction is category *j*, and $q\left(x_{ij}\right)$ denotes the label that is actually category j. Usually, $q\left(x_{ij}\right)=1$ if it belongs to category j, and $q\left(x_{ij}\right)=0$ if it does not.

The second branch guided by the orange arrow implements the estimation of the DOA for each source; the activation function of this branch is the Leaky ReLU function. This branch is a regression problem; its loss function is the mean squared difference loss function, whose expression is:(14)$$L\left(y,y^{\prime }\right)=\frac{\sum_{i=1}^{M}\sum_{j=1}^{N}{\left(y_{ij}-y_{ij}^{\prime }\right)}^{2}}{MN},$$
where M denotes the number of samples, N denotes the number of outputs, $y_{ij}$ denotes the true value, and $y_{ij}^{\prime }$ denotes the output value, i.e., DOA estimation in this paper.

The two branches are independent of each other, but neurons of the first two layers are jointly influenced by the two branches.

## 4. Simulation Experiments and Analysis of Results

In the simulation experiments, an ideal ULA and a sparse array with eight array elements are designed, where the array elements of ULA are arranged as $\left[0,1,2,3,4,5,6,7,\right]\cdot \left(\lambda /2\right)$, the sparse array is a co-prime array with co-prime numbers of 4 and 5, whose array elements are arranged as $\left[0,4,5,8,10,12,15,16\right]\cdot \left(\lambda /2\right)$, and λ is the signal wavelength. The source number is increased from 1 to 18, and the target region is $\left[-60{}^{\circ},60{}^{\circ}\right]$. In the proposed model, there are three convolutional layers, the size of convolutional kernels is 3×3, and the number of convolutional kernels is 64, 64 and 32. The fully connected layers are designed as shown in Figure 2; the number of neurons in the first two layers is 1000 and 800, and the probability of dropout is 0.2. The probabilities of dropout are 0.2 and 0.1 for Output1, and the probabilities of dropout are 0.3 and 0.2 for Output2, which has two fully connected layers with 800 and 600 neurons. Since a series of operations such as convolution operations and activation functions in the forward and backward propagation processes in CNN are all real operations, and the model outputs are also real numbers, the complex components are not suitable for participation in them, but the complex components are still meaningful for angle estimation and cannot be discarded. However, the complex components remain significant for angle estimation and cannot be ignored. In the simulation, the real and imaginary components of the covariance matrix are thus split apart and combined to create a M×2M matrix (M indicates the actual number of elements), which is used as the convolutional neural network’s input. The total number of training sets is 36,000, and the total number of test sets is 5400. The simulation experiments are designed to implement source number estimation and DOA estimation when the number of sources is unknown.

### 4.1. Performance of Source Number Estimation

The accuracy and consistency rates are used as indicators for evaluating the performance of the source number estimation, where the accuracy rate indicates the proportion of the source number estimate that is the same as the actual value; it is used to evaluate the good or bad performance of the model for the source number estimation. The expressions are:(15)$$P_{a}=\frac{N_{a}}{N_{t}},$$
where $N_{a}$ denotes the number of accurate estimates achieved in the test set, and $N_{t}$ denotes the test set capacity.

Although Output2 outputs the target’s DOA, it is also able to reflect the number of sources from it. The consistency rate represents the proportion of the test set in which the estimated number of sources output by Output1 and the number of sources obtained from Output2 are the same; it is used to measure the degree of consistency between the two output branches and is expressed as:(16)$$P_{c}=\frac{N_{c}}{N_{t}},$$
where $N_{c}$ denotes the amount of data with the same number of sources obtained by Output1 and Output2. Statistically, the accuracy of the source number estimation for different numbers of sources received by the model for ULA and sparse array at the snapshot number of 200 and different SNRs are shown in the table below Table 2 and Table 3.

From Table 2 and Table 3, it can be seen that the accuracy of estimation for a different number of sources for both sparse arrays and ULA is above 90% at different SNRs. When there are no more than 12 sources for sparse arrays and no more than 6 sources for ULA, both are capable of achieving an accuracy of 100%. However, as the number of sources increases, it is evident that the estimation accuracy under high SNR conditions is higher. Nearly all estimation errors are +1 or −1 for situations where the estimated number of sources is different from the actual number of sources. Consistency of the source number estimation for different numbers of sources received by the model for ULA and sparse array at the snapshot number of 200 and different SNRs are shown in the table below Table 4 and Table 5.

Comparing Table 4 and Table 5, it can be seen that the model estimates the number of sources for sparse arrays slightly more accurately than it does for ULA. Comparing Table 2, Table 3, Table 4 and Table 5, it can be seen that the accuracy and consistency rate of the source number estimation are essentially the same, with a slightly lower consistency rate when the source number increases and the SNR is small. As a result, the more precise performance of Output2’s source number is marginally inferior to Output1’s. In particular, it is possible that some angles cannot be estimated when two or more sources have similar angles of incidence, which could result in underreporting.

### 4.2. Performance of DOA Estimation

When DOA estimation of Output2 has a large deviation from the target region, it is regarded as an invalid estimation. In addition, all the filler values of N set in the text are invalid estimations. Usually, the root mean square error (RMSE) is used as an index to evaluate the accuracy of DOA estimation, and its expression is:(17)$$\mathrm{RMSE}=\sqrt{\frac{\sum_{i=1}^{N_{t}}{\left({\tilde{\theta }}_{i}-\theta_{i}\right)}^{2}}{N_{t}}},$$
where $N_{t}$ denotes the number of all valid sources, ${\tilde{\theta }}_{i}$ denotes the estimated angular value, and $\theta_{i}$ denotes the actual angular value.

#### 4.2.1. DOA Estimation Performance at Different SNRs

When the number of snapshots is 200 and SNRs are −5 dB, 0 dB, 5 dB, 10 dB, 15 dB, 20 dB, 25 dB, ULA and the sparse array perform DOA estimation is conducted when the source number is unknown to obtain Output2; the RMSE of DOA estimation at different SNRs with the number of sources from 1 to 18 respectively is shown in the Table 6 and Table 7.

Table 6 and Table 7 show that for both ULA and sparse array, estimation accuracy increases with increasing SNR while decreasing with increasing sources. Particularly, estimation accuracy decreases more noticeably for sparse array when the number of sources is greater than 16 and for ULA when the number of sources is greater than 6. With the same number of sources and SNR as shown in Table 6 and Table 7, sparse arrays’ estimation accuracy is marginally better than the ULA’s. Regardless of the quantity of sources, they perform better during estimation for sparse arrays when the SNR is higher than 10 dB RMSE of DOA estimation for ULA and sparse arrays for all valid sources (sources numbered from 1 to 18) at the same SNR. Comparing the RMSE of the algorithm proposed in this paper with the differential co-array joint MUSIC algorithm (DCAM) [27], L1SVD algorithm [28], and L1CMSR algorithm [29] for sparse arrays of the same array type, for ULA, the MUSIC algorithm [30] is used for multi-objective DOA estimation with the same array type. Only consecutive differential joint array elements can be used in the simulation experiments of this paper when using the differential common array algorithm. According to the co-prime array mentioned in this paper, its maximum number of consecutive array elements is 13, so the maximum number of sources that can be estimated by this algorithm is 12. The L1SVD algorithm and the L1CMSR algorithm can estimate a maximum of 16 sources each. Therefore, the number of sources for the aforementioned three algorithms is planned to be 1 to 12, 1 to 16, and 1 to 16 in the simulation experiments, while the MUSIC algorithm’s maximum number of estimable sources must be less than the number of array elements, or 7 in this case. As a result, 1 to 7 sources are intended to be used in the comparison experiments for the MUSIC to perform DOA estimation on ULA. The results are shown in Figure 4.

As can be seen from Figure 4, each algorithm’s RMSE decreases and estimation accuracy increases as the SNR rises. When the array is a sparse array, the algorithm suggested in this paper performs better than DCAM, L1SVD, and L1CMSR. When the array is a ULA, the algorithm outperforms MUSIC when the SNR is less than 5 dB, and the MUSIC algorithm performs better when the SNR is greater than 5 dB. It should be noted that the other four algorithms require a known number of sources to achieve DOA estimation, and the MUSIC algorithm is prone to generating false spectral peaks or no spectral peaks when the source number is large and the angles of incidence of the sources are similar, i.e., when the angular difference is small, which affects the estimation performance.

#### 4.2.2. DOA Estimation Performance at Different Snapshots

When the number of sources is unknown, the SNR is 10 dB and the number of snapshots is 50, 100, 150, 200, 300, and 400, respectively, the ULA and sparse array perform DOA estimation to obtain Output2. According to statistics, Table 8 and Table 9 below show the estimated RMSE of DOA at various snapshots, with the source number ranging from 1 to 18, respectively.

From Table 8 and Table 9, it can be seen that the estimation accuracy rises with the number of snapshots and that the estimation error is higher when the number of sources is high and the number of snapshots is small. When there are many sources, there is a greater chance that there will be two or more sources with a close angle of incidence. When this occurs, the angle estimation error and RMSE both rise along with the number of sources.

Under the above conditions, the RMSEs of the proposed algorithm for DOA estimation of sparse arrays and ULAs for all valid sources (number of sources from 1 to 18) at the same snapshot are calculated. For sparse arrays, the algorithm is compared with the DCAM [20], L1SVD [21], and L1CMSR [22] algorithms for the RMSE of DOA estimation under the same conditions, and with the MUSIC [23] algorithm for RMSE comparison for ULA. The maximum number of measurable sources for each algorithm is consistent with the above. The results are shown in Figure 5.

From Figure 5, it is clear that as the number of snapshots rises, each algorithm’s estimation accuracy gets better. When the array is ULA, it can be seen that when the number of snapshots is less than 200, the RMSE of the proposed algorithm is significantly lower than that of MUSIC algorithm, and the estimation accuracy of the proposed algorithm is higher, while when the number of snapshots is greater than 200, the estimation accuracy of the two algorithms is not significantly different. The estimation accuracy of the proposed algorithm is superior to the other three algorithms when the array is sparse, and its RMSE is also lower. When working with sparse arrays and ULA, it is clear from comparing Figure 4 and Figure 5 that the SNR has a greater impact on the algorithm than the number of snapshots.

#### 4.2.3. Performance at Small Snapshots and Low SNR

From Table 6, Table 7 and Table 8, as well as Figure 3 and Figure 4, it can be seen that both the number of snapshots and the SNR affect the estimation accuracy of the source number and the RMSE of the DOA estimation. To verify the estimation performance of the algorithm proposed in this paper under the conditions of low SNR and small snapshots, the ULA and sparse array are designed to perform source number and DOA estimation under the conditions of a snapshot number of 50 and SNR of −5 dB; the accuracy of source number estimation and RMSE of DOA estimation under each source number is shown in Figure 6.

It can be seen from Figure 6a that the accuracy of source number estimation de-creases as the source number increases at low snapshot and low SNR. When the source number does not exceed 8, for sparse arrays and ULA, the accuracy is close to 100%; when it is greater than 8, the accuracy of source number estimation for sparse arrays is significantly better than ULA. After the number of sources exceeds 12, the accuracy of the source number for ULA estimation decreases significantly; when the number of sources reaches 18, and the accuracy of the estimation is still greater than 80%, while the accuracy of the estimation of the source number for the sparse array decreases significantly when the source number is greater than 15, and the accuracy is more than 85% when the source number reaches 18. As can be seen from Figure 6b, the RMSE of both the sparse array and ULA increases with the source. The DOA estimation performance of the sparse array is significantly better than that of ULA, and the DOA estimated RMSE of ULA increases significantly when the number of sources is greater than six; in particular, when the source number is greater than 15, the estimated RMSE exceeds 1°, the RMSE of the sparse array is less than 0.6° when the number of sources is less than 9, and the RMSE is around 0.7° when the number of sources is greater than 13. Combined with Figure 6a,b, it can be seen that the proposed algorithm still has good performance in source number estimation and DOA estimation at low snapshot and low SNR and can meet the practical requirements.

## 5. Discussion

In the field of array signal processing, the source number is usually a prerequisite for estimating signal-related parameters. In order to simplify the source number and DOA estimation problem and improve the estimation accuracy, a joint source number and DOA estimation method based on an improved convolutional neural network is proposed in this paper. The model discards the pooling layer in the traditional convolutional neural network to fully retain and extract the features of the covariance matrix and introduces the dropout method in the fully connected layer to improve the generalization ability of the model. Simulation experiments show that the algorithm is not restricted by the type of array and can achieve joint estimation of source number and DOA for both sparse arrays and ULAs with high estimation accuracy. When the number of sources is large, the classical algorithm is difficult to apply, but the algorithm in this paper still has high estimation accuracy; especially under low signal-to-noise ratio and small snapshot conditions, the algorithm still has high estimation accuracy. The analysis of the model and results shows that in the signal model, the larger the number of snapshots, the closer the sampled signal is to the real signal and the more accurate the feature extraction is, while the higher the signal-to-noise ratio, the less the signal is disturbed by noise and the easier it is to extract effective features. Thus, in both cases, the angle estimation of the algorithm proposed in this paper is higher, and as the signal-to-noise ratio and the number of snapshots further increase, the difference between the algorithm proposed in this paper and the traditional algorithm gradually decreases. On the one hand, the traditional algorithm is more directly influenced by the signal-to-noise ratio and snapshot; on the other hand, it is limited by the fact that when the covariance matrix is used as input, it is not a complete signal covariance matrix, but a matrix of N×2N with the real and imaginary parts stitched together to facilitate model training. This will also affect the feature extraction to a certain extent, so there is a limitation in the estimation accuracy. For the case of low SNR and small snapshots, the network model is estimated by extracting and combining features layer by layer, which is less affected by low SNR and small snapshots than the traditional algorithm based on feature decomposition for angle estimation; thus, the estimation accuracy is higher than the traditional algorithm under the condition of low SNR and small snapshots; combined with the limitation of the array’s own degrees of freedom, the performance of the proposed algorithm degrades significantly after the number of sources exceeds a certain range. On the other hand, when the number of sources is large, the possibility of a smaller angular interval increases, and when the angular interval is small, the estimation error is likely to increase. Thus, in the application, when the spatial target is dense or the number of sources is too large, the algorithm proposed in this paper is suitable as an auxiliary reference. In future research, making fuller use of the signal covariance matrix in order to extract features more effectively for higher accuracy DOA estimation becomes one of the research priorities.

## Figures and Tables

**Figure 1 sensors-23-03100-f001:**
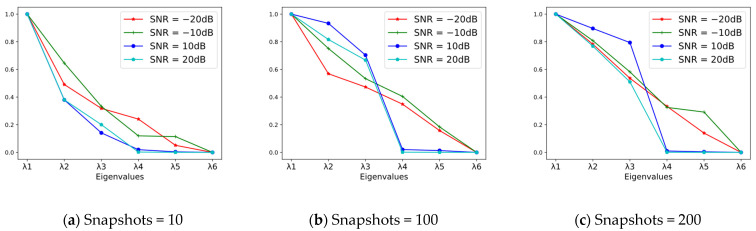
Distribution of eigenvalues after normalization. ((**a**) Snapshots = 10; (**b**) Snapshots = 100; (**c**) Snapshots = 200).

**Figure 2 sensors-23-03100-f002:**
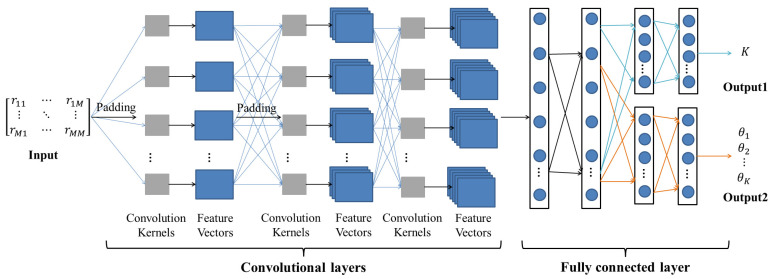
The structure of the CNN model.

**Figure 3 sensors-23-03100-f003:**
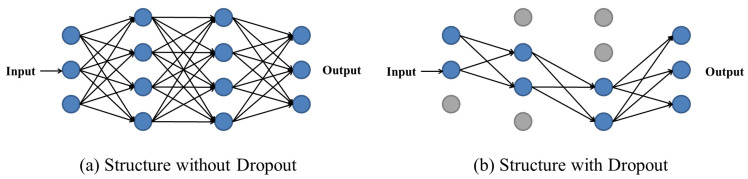
Illustration of the dropout method. ((**a**) Structure with = out Dropout; (**b**) Structure with Dropout).

**Figure 4 sensors-23-03100-f004:**
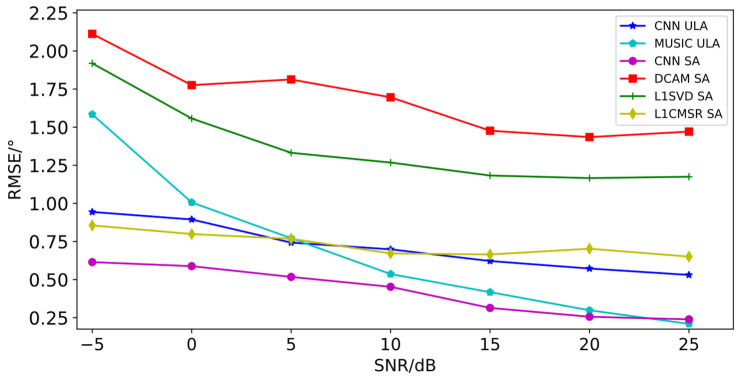
Comparison of RMSE of various algorithms with different SNRs.

**Figure 5 sensors-23-03100-f005:**
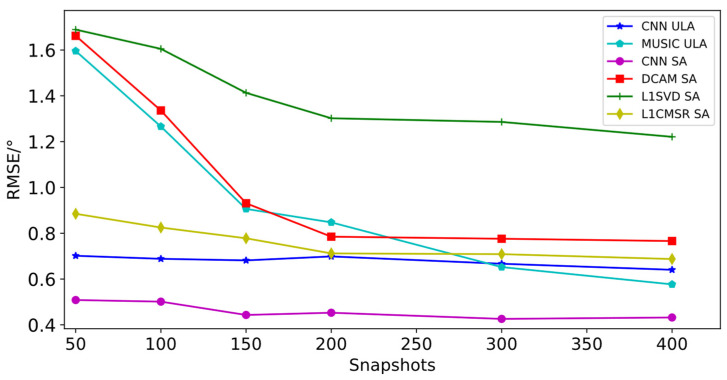
Comparison of RMSE of various algorithms with different snapshots.

**Figure 6 sensors-23-03100-f006:**
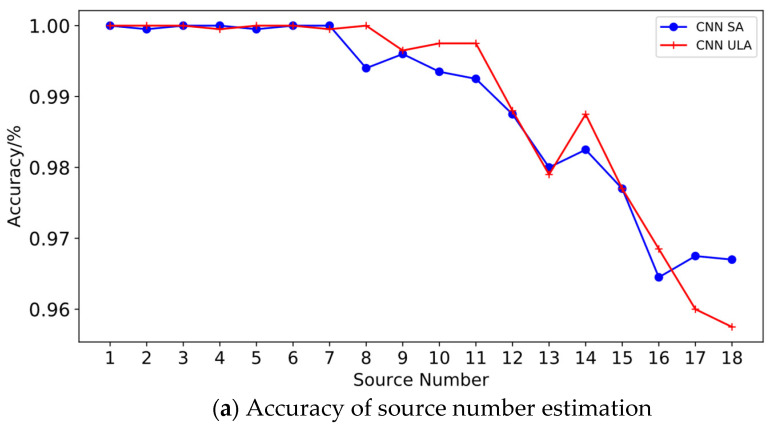
Performance of CNN algorithms for source number and DOA estimation at low snapshot and low SNR.

**Table 1 sensors-23-03100-t001:** Eigenvalues at different snapshots and SNR.

Snapshots	10				100				200			
SNR/dB	−20	−10	10	20	−20	−10	10	20	−20	−10	10	20
$\lambda_{1}$	954.39	74.69	11.19	8.54	392.34	40.62	7.56	7.21	376.96	41.73	6.42	8.53
$\lambda_{2}$	504.65	49.74	4.27	3.25	307.01	34.66	7.07	5.89	345.08	37.99	5.78	6.56
$\lambda_{3}$	352.02	27.73	1.61	1.72	288.04	29.45	5.39	4.82	309.06	33.59	5.15	4.36
$\lambda_{4}$	283.77	12.69	0.255	0.029	263.47	26.37	0.393	0.033	279.27	28.59	0.314	0.034
$\lambda_{5}$	116.21	12.31	0.081	0.012	225.69	21.09	0.340	0.027	251.00	27.93	0.278	0.028
$\lambda_{6}$	70.92	4.26	0.034	0.008	194.48	16.68	0.242	0.022	230.54	22.25	0.251	0.026

**Table 2 sensors-23-03100-t002:** Accuracy of ULA source number estimation at different SNRs.

SNR/dB		−5	0	5	10	15	20	25
Source Number	1–6	100	100	100	100	100	100	100
	7	100	99.85	100	100	100	100	100
	8	99.55	100	99.80	100	99.95	100	100
	9	100	100	100	100	100	100	100
	10	99.80	100	98.50	100	100	100	100
	11	100	100	100	99.90	100	99.85	99.95
	12	99.50	98.65	100	100	100	100	99.90
	13	99.80	100	100	99.95	99.75	99.95	100
	14	98.95	99.85	99.90	99.75	99.50	99.85	99.95
	15	97.90	98.65	98.45	99.50	99.80	99.70	99.60
	16	96.10	96.15	97.20	98.50	98.60	99.45	98.75
	17	95.10	96.20	95.80	97.45	97.20	96.15	97.00
	18	94.15	95.55	95.75	97.65	97.95	98.80	98.50

**Table 3 sensors-23-03100-t003:** Accuracy of sparse array source number estimation at different SNRs.

SNR/dB		−5	0	5	10	15	20	25
Source Number	1–12	100	100	100	100	100	100	100
	13	99.55	98.70	99.75	98.90	100	98.95	100
	14	98.75	99.45	99.85	98.45	99.65	99.70	100
	15	99.10	100	99.35	100	98.90	99.45	97.70
	16	97.85	99.05	99.60	99.85	98.75	100	99.05
	17	97.50	99.45	99.10	97.5	99.60	98.30	97.55
	18	96.35	97.80	98.50	97.65	98.30	99.20	99.25

**Table 4 sensors-23-03100-t004:** Consistency rates of ULA source number estimation at different SNRs.

SNR/dB		−5	0	5	10	15	20	25
Source Number	1–6	100	100	100	100	100	100	100
	7	99.95	99.85	100	99.90	100	99.95	100
	8	99.50	100	99.70	100	99.85	100	99.95
	9	99.85	99.5	99.75	99.95	100	99.9	100
	10	99.75	100	98.25	100	99.80	100	99.90
	11	98.95	99.85	100	99.90	100	99.85	99.85
	12	99.50	98.65	99.80	100	99.85	100	99.90
	13	99.35	98.95	99.25	99.95	99.75	99.95	100
	14	98.75	99.85	99.90	99.70	99.35	99.25	99.65
	15	97.40	98.60	98.45	99.35	99.70	99.15	98.95
	16	95.85	96.15	97.20	98.45	98.50	99.05	98.55
	17	94.35	96.15	95.25	96.35	96.80	95.95	96.75
	18	94.05	93.95	94.25	97.35	97.95	97.80	98.50

**Table 5 sensors-23-03100-t005:** Consistency rates of sparse array source number estimation at different SNRs.

SNR/dB		−5	0	5	10	15	20	25
Source Number	1–4	100	100	100	100	100	100	100
	5	99.95	100	100	100	100	100	100
	6	100	100	100	100	99.95	100	100
	7	100	100	100	100	100	100	100
	8	99.80	100	99.95	100	100	100	100
	9–11	100	100	100	100	100	100	100
	12	100	99.95	100	100	99.90	99.85	100
	13	99.30	98.65	99.75	98.90	100	98.95	99.95
	14	98.65	99.40	99.65	98.45	99.60	99.65	100
	15	99.05	99.85	99.35	99.80	98.75	99.25	97.70
	16	97.45	98.65	99.55	99.55	98.70	99.80	99.00
	17	96.95	98.85	98.55	96.75	99.55	98.20	97.55
	18	96.35	97.65	98.35	96.35	98.25	99.10	98.75

**Table 6 sensors-23-03100-t006:** RMSE of DOA estimation for ULA at different SNRs.

SNR/dB		−5	0	5	10	15	20	25
Source Number	1	0.6425	0.4169	0.3302	0.2185	0.1674	0.1127	0.0907
	2	0.5209	0.4912	0.3750	0.3220	0.3163	0.2006	0.1740
	3	0.5391	0.4136	0.4128	0.3718	0.3293	0.3215	0.2043
	4	0.5270	0.4096	0.3521	0.3649	0.2546	0.2500	0.1748
	5	0.5446	0.5384	0.3675	0.3661	0.3545	0.3299	0.2709
	6	0.6138	0.5912	0.4588	0.4214	0.3454	0.3189	0.3024
	7	0.6939	0.6600	0.6225	0.5676	0.4491	0.3357	0.3012
	8	0.6884	0.6841	0.6407	0.6945	0.5704	0.4837	0.4021
	9	0.7293	0.7011	0.6564	0.6316	0.5584	0.5528	0.5328
	10	0.7980	0.7077	0.6559	0.6423	0.5883	0.5705	0.5547
	11	0.7724	0.7637	0.6968	0.6171	0.5878	0.5720	0.5702
	12	0.7685	0.7960	0.7530	0.6335	0.5889	0.5708	0.5671
	13	0.8914	0.7206	0.7452	0.6205	0.6179	0.6024	0.5157
	14	1.0573	1.0718	0.6138	0.6543	0.6381	0.6778	0.5631
	15	1.0854	1.0813	0.8528	0.7369	0.6540	0.5817	0.5232
	16	1.1623	1.0831	0.8521	0.9402	0.7909	0.6080	0.6010
	17	1.1949	1.1353	0.8200	0.7391	0.6403	0.6120	0.5941
	18	1.2056	1.1018	1.0360	0.9461	0.8307	0.7500	0.6928

**Table 7 sensors-23-03100-t007:** RMSE of DOA estimation for sparse array at different SNRs.

SNR/dB		−5	0	5	10	15	20	25
Number of Sources	1	0.6138	0.3341	0.2283	0.1974	0.1127	0.0905	0.0378
	2	0.5446	0.3735	0.2211	0.1740	0.2032	0.0985	0.0780
	3	0.5270	0.3634	0.3451	0.2895	0.2325	0.0963	0.0907
	4	0.4281	0.5442	0.4516	0.3850	0.2207	0.1067	0.1253
	5	0.4505	0.6416	0.2404	0.2404	0.2695	0.1716	0.0929
	6	0.4196	0.3619	0.3619	0.1439	0.2688	0.1622	0.0781
	7	0.6338	0.5979	0.5026	0.3474	0.3101	0.2426	0.1674
	8	0.7169	0.7145	0.6007	0.3091	0.2309	0.2050	0.1644
	9	0.4597	0.5659	0.5016	0.3904	0.3407	0.2031	0.1804
	10	0.6451	0.5151	0.3921	0.3471	0.2517	0.2266	0.2051
	11	0.4934	0.5270	0.3187	0.3044	0.2613	0.2319	0.2259
	12	0.4799	0.4779	0.3742	0.3304	0.2599	0.2250	0.2161
	13	0.6254	0.4938	0.4790	0.4500	0.3632	0.2493	0.2251
	14	0.6504	0.6287	0.6230	0.5788	0.3454	0.3189	0.3024
	15	0.5746	0.5077	0.4919	0.4609	0.3567	0.2625	0.3061
	16	0.6453	0.6357	0.5637	0.4944	0.3596	0.2990	0.2886
	17	0.8004	0.7737	0.7045	0.6517	0.3620	0.3094	0.2753
	18	0.6888	0.6501	0.6284	0.5805	0.3204	0.3177	0.2917

**Table 8 sensors-23-03100-t008:** RMSE of DOA estimation for ULA at different snapshots.

Snapshots		50	100	150	200	300	400
Source Number	1	0.3241	0.3016	0.2740	0.2185	0.2302	0.2035
	2	0.3776	0.3444	0.3090	0.3220	0.3131	0.3326
	3	0.4087	0.3746	0.3209	0.3718	0.3379	0.3269
	4	0.4552	0.4055	0.3492	0.3649	0.3017	0.3006
	5	0.4259	0.4569	0.4079	0.3661	0.3621	0.3421
	6	0.4706	0.4745	0.4889	0.4214	0.4023	0.4128
	7	0.5822	0.5809	0.5811	0.5676	0.5255	0.5490
	8	0.6359	0.6272	0.5445	0.6945	0.6260	0.5650
	9	0.5962	0.6572	0.6175	0.6316	0.6604	0.5792
	10	0.6271	0.6019	0.6524	0.6423	0.5948	0.6110
	11	0.6485	0.6231	0.6450	0.6171	0.6055	0.5781
	12	0.6705	0.6606	0.6437	0.6335	0.6403	0.6071
	13	0.6687	0.6577	0.6477	0.6205	0.6490	0.6139
	14	0.6891	0.6757	0.6524	0.6543	0.6313	0.5903
	15	0.7057	0.7652	0.7573	0.7369	0.6696	0.6470
	16	0.7625	0.7913	0.8045	0.9402	0.7633	0.8027
	17	0.8431	0.8096	0.7924	0.7391	0.7916	0.7405
	18	0.9738	0.8668	0.8786	0.9461	0.9148	0.8556

**Table 9 sensors-23-03100-t009:** RMSE of DOA estimation for sparse array at different snapshots.

Snapshots		50	100	150	200	300	400
Source Number	1	0.2295	0.2051	0.1416	0.1974	0.1907	0.2127
	2	0.2303	0.3010	0.2298	0.1740	0.1569	0.1860
	3	0.3520	0.3996	0.3238	0.2895	0.3013	0.2170
	4	0.3321	0.3458	0.3102	0.3850	0.3552	0.3108
	5	0.3790	0.3649	0.3001	0.2404	0.3125	0.2375
	6	0.3376	0.2582	0.2765	0.1439	0.1710	0.2067
	7	0.3757	0.2327	0.3300	0.3474	0.3232	0.2814
	8	0.3548	0.3405	0.3365	0.3091	0.2810	0.2613
	9	0.3734	0.4020	0.3892	0.3904	0.3365	0.3647
	10	0.3813	0.4181	0.3363	0.3471	0.3300	0.3010
	11	0.3893	0.3517	0.3188	0.3044	0.3087	0.2824
	12	0.4166	0.4048	0.3494	0.3304	0.3255	0.4014
	13	0.4615	0.5273	0.4631	0.4500	0.4117	0.3801
	14	0.5974	0.5716	0.4605	0.5788	0.5314	0.4662
	15	0.5608	0.5760	0.4901	0.4609	0.4586	0.4866
	16	0.5742	0.5578	0.4844	0.4944	0.4786	0.5428
	17	0.6907	0.6767	0.6005	0.6517	0.5466	0.5711
	18	0.6717	0.6486	0.5933	0.5805	0.5816	0.6027

## Data Availability

The data presented in this study are available on request from the corresponding author. The data are not publicly available, due to the data in this paper not being from publicly available datasets but obtained from the simulation of the signal models listed in the article.

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
