# Peer review of "Research on Underdetermined DOA Estimation Method with Unknown Number of Sources Based on Improved CNN"

_sensors, 2023, doi:10.3390/s23063100_

Round 1

Reviewer 1 Report

Research on Underdetermined DOA Estimation Method with Unknown Number of Sources Based on Improved Deep Convolutional Neural Network

In this study, an example of the deep learning method for DOA estimation is presented. CNN structure construction method for the DOA estimation which is a changing output-sized problem is described.But the following points need to be handled.

1- There are many singular and plural usage errors in the English of the text. The entire text should be carefully reviewed for grammar.
2- It is said "on Improved Deep Convolutional Neural Network" in the title. Here you should clearly state how you made the improvement in DCNN.
3- In equation 1, a(Theta_k) is used. but in line 132 a(Theta_i) is used. For the same equations, same index (k instead of i) need to be used.
4- In Table 1, what the Snapshot/Times means? Does it mean number of collected samples (or snapshots)? Please search in literature for correct useage.
5- In Table 1, what is the meaning of SNR/dB.
The values in Table 1 could be presented also with a graphic (by normalizing Lambda values to 1) to make comparison easy.
6- In line 273, it is presented that "y'ij' denotes the output value". Does the "output value" mean estimated value?
7- In Table 2, 3, 4, 5, 6, 7, 8, 9  what are the values in first column?
8- The quality of figure 3, 4, 5 needs to be improved.
9- What is the antenna count of the ULA used in all submitted test results? (in source number estimation and angle estimation tests) Does it change or is it constant?
10- How was the dataset obtained? Were they measurement or simulation results?
11- A few samples of dataset records could be shown in a table.
12- There is no information about the complex components of the covariance matrix. The methods like MUSIC, ESPRIT, Minimum variance distortionless response etc. also use the complex components of data collected from antennas. For this purpose, there are various methods to calculate the complex components of a vector. Do you use the complex components of the data collected from the antenna array? If so, how did you employ the CNN with complex-valued matrice?

Reviewer 2 Report

Please extends your algorithms to 2D arrays. Eliminates tablets with dará,bits prefereable inly hraphs or move to an anexa. 

Author Response

Thank you for your suggestion, the simulation results show that the current algorithm does not work perfectly for 2D arrays and we are currently working on new models to apply to 2D arrays.

Reviewer 3 Report

This paper proposes a joint source number and underdetermined DOA estimation method based on an improved convolutional neural network. The work contains various issues which are listed as follow

1. The title must be simplified and understandable, what does it means by the word deep CNN? If the authors can us DOA word in the title then they should use CNN word instead of defined form, because CNN is known to all computer electrical and electronics related researchers. 

2. The abstract of the paper is very poor,  the authors should first study about CNN, and DOA. There is no need to discuss in the abstract the "The neural network model is a deep convolutional neural network, and the joint estimation of 14 source number and DOA is achieved simultaneously through data training." They should write about the methodology, problem and the achieved results in the abstract. 

3.  In the abstract the authors have written that they have used DCNN and in the keywrods the CNN is mentioned. 

4. Figure 1 is about CNN not DCNN. 

5. The proposed model is not included in the work, The results discussion are very poor,.

6. In conclusion the complete presentation of the work is very poor, the explanation of abbreviations are not as per standard format.  Very old literature is studied,  no current is investigated in the paper. The authors must have to revised complete manuscript in terms current problems, proposed model, used flowchart, mathematical calculations and simulation model. 

Round 2

Reviewer 3 Report

The comments are addressed.